# Extracellular Vesicles Potentiate Medulloblastoma Metastasis in an EMMPRIN and MMP-2 Dependent Manner

**DOI:** 10.3390/cancers15092601

**Published:** 2023-05-04

**Authors:** Hannah K. Jackson, Christine Mitoko, Franziska Linke, Donald Macarthur, Ian D. Kerr, Beth Coyle

**Affiliations:** 1Children’s Brain Tumour Research Centre, School of Medicine, Biodiscovery Institute, University Park, University of Nottingham, Nottingham NG7 2RD, UK; hj363@cam.ac.uk (H.K.J.); f.linke@erasmusmc.nl (F.L.); 2Department of Pathology, University of Cambridge, Cambridge CB2 1QP, UK; 3Department of Experimental Urology, Erasmus MC Cancer Institute, University Medical Center Rotterdam, 3015 GD Rotterdam, The Netherlands; 4Department of Neurosurgery, Nottingham University Hospital, Nottingham NG7 2UH, UK; 5School of Life Sciences, University of Nottingham, Queen’s Medical Centre, Nottingham NG7 2UH, UK; ian.kerr@nottingham.ac.uk

**Keywords:** medulloblastoma, extracellular vesicles, exosomes, MMP-2, EMMPRIN, metastasis

## Abstract

**Simple Summary:**

Medulloblastoma is the most prevalent malignant paediatric brain tumour, where metastasis and recurrence account for 95% of medulloblastoma-associated deaths. Secretion of extracellular vesicles (EVs) has emerged as a pivotal mediator for communication in the tumour microenvironment during metastasis. We investigated whether sEVs and exosomes mediate communication between medulloblastoma cells and their surroundings to drive metastasis. Metastatic exosomes were shown to potentiate medulloblastoma migration via the active protease, matrix metalloproteinase-2 (MMP-2), on their surface, resulting in degradation of the extracellular matrix (ECM) and creating routes for medulloblastoma cells to invade into the surrounding environment. Knockdown of MMP-2 and its activator extracellular matrix metalloproteinase inducer (EMMPRIN) reduced this invasive potential. Our observations also highlight the potential of MMP-2 as a biomarker for metastatic medulloblastoma. Together, our findings reveal unique insights into the pathogenesis of medulloblastoma and highlight the need to explore alternative therapeutic approaches to impair MMP-driven mechanisms of tumour invasion and migration.

**Abstract:**

Extracellular vesicles (EVs) have emerged as pivotal mediators of communication in the tumour microenvironment. More specifically, nanosized extracellular vesicles termed exosomes have been shown to contribute to the establishment of a premetastatic niche. Here, we sought to determine what role exosomes play in medulloblastoma (MB) progression and elucidate the underlying mechanisms. Metastatic MB cells (D458 and CHLA-01R) were found to secrete markedly more exosomes compared to their nonmetastatic, primary counterparts (D425 and CHLA-01). In addition, metastatic cell-derived exosomes significantly enhanced the migration and invasiveness of primary MB cells in transwell migration assays. Protease microarray analysis identified that matrix metalloproteinase-2 (MMP-2) was enriched in metastatic cells, and zymography and flow cytometry assays of metastatic exosomes demonstrated higher levels of functionally active MMP-2 on their external surface. Stable genetic knockdown of MMP-2 or extracellular matrix metalloproteinase inducer (EMMPRIN) in metastatic MB cells resulted in the loss of this promigratory effect. Analysis of serial patient cerebrospinal fluid (CSF) samples showed an increase in MMP-2 activity in three out of four patients as the tumour progressed. This study demonstrates the importance of EMMPRIN and MMP-2-associated exosomes in creating a favourable environment to drive medulloblastoma metastasis via extracellular matrix signalling.

## 1. Introduction

Metastatic medulloblastoma is a devastating disease with a poor prognosis of less than 10% five-year survival in affected paediatric patients [1,2]. Whilst primary medulloblastoma tumours have been well characterised on the basis of their epigenetic and transcriptomic features, there is very little information regarding the molecular signatures of metastatic tumours due to the rarity of reoperation. The resulting scarcity of metastatic tumour samples from these patients limits data available on the metastatic tumour microenvironment, which has been increasingly postulated to exert a role in the dissemination of solid tumours.

The secretion of extracellular vesicles (EVs) as vehicles for cell-to-cell communication has gained prominence in recent years. The most common nomenclature divides EVs into three subclasses based on their biogenesis; exosomes, approximately 30–150 nm in diameter, originating in multivesicular endosomes (MVEs) which fuse with the plasma membrane; microvesicles, which range from 100–1000 nm and form via direct budding of the plasma membrane; and apoptotic bodies, vesicles ranging around 1–4 µm and are formed from of dying cells [3]. The most investigated are small EVs (sEVs or exosomes), which are nanometer-sized vesicles secreted by all cell types and able to cross the blood–brain barrier (BBB) [4]. The role of exosomes as vehicles for cell-to-cell communication between a tumour and its microenvironment is a relatively new concept, with only limited study of their role in medulloblastoma [5,6,7,8,9]. Exosomes represent a unique form of information delivery operating at short and long distances [10]. Tumour-derived exosomes can transfer signals and convey information from tumours to distant tissues and organs. They are also present in circulation and can therefore disseminate their cargo throughout the body. Exosomes have been shown to carry surface components that enable direct contact with recipient cells to activate intracellular signalling [11,12,13]. In addition, exosomes interact with target cells by fusion with the cell membrane followed by the transfer of exosomal cargoes (protein, mRNA, miRNA) into the cell cytoplasm [14]. Cancer cell-derived exosomes have been shown to participate in the crucial steps of the metastatic spread of a primary tumour, ranging from oncogenic reprogramming of malignant cells to the formation of premetastatic niches [15]. These effects are achieved through the mediation of intercellular cross talk and subsequent modification of both local and distant microenvironments. In the context of brain tumours, sEVs originating from primary tumour cells have been demonstrated to enhance the growth of brain metastases, modulate immune responses to support tumour cell proliferation, regulate tumour cell stability, and have been identified as potential diagnostic or prognostic biomarkers [10,16,17]. This study focussed on how medulloblastoma exosomes interact with the recipient cells utilizing direct extracellular matrix signalling, via surface-associated proteins, to activate intracellular signalling pathways.

In this context, we assessed the expression of tumour-supporting proteins EMMPRIN and MMP-2. Increased levels of MMP-2 and EMMPRIN have been associated with more aggressive metastatic disease in other solid tumours [18,19]. Therefore, we sought to investigate whether the secretion of exosomes containing MMPs and EMMPRIN might be necessary for tumour progression and thereby a potential mechanism of dissemination in medulloblastoma. We demonstrated that metastatic exosomes significantly enhanced the migration and invasiveness of primary and nonmalignant cells. Moreover, the promigratory function of metastatic exosomes was, in part, due to EMMPRIN and MMP-2 enriched on their external surface. Furthermore, metastatic exosomes were shown to potentiate medulloblastoma migration resulting in the degradation of the extracellular matrix components, via the active protease, MMP-2, on their surface. In support of this, stable genetic knockdown of the genes encoding MMP-2 and EMMPRIN (namely *MMP-2* or *BSG*) antagonised the promigratory function of exosomes, confirming that MMP-2 and EMMPRIN are promigratory factors on medulloblastoma exosomes. Finally, we observed increased levels of functionally-active MMP-2 in medulloblastoma patients’ CSFs, which correlated with disease progression and appeared to correlate with prognosis, highlighting the biomarker potential of functional MMP-2.

## 2. Materials and Methods

### 2.1. Cell Culture

CHLA-01 and CHLA-01R brain tumour cell lines (primary-tumour-derived cell line and recurrent metastatic line derived from pleural fluid, respectively) were obtained from Geoff Pilkington (University of Portsmouth, Portsmouth, UK). DAOY (nonmetastatic primary-tumour derived) and D283 (derived from peritoneal metastases) brain tumour cell lines were obtained from ATCC (Manassas, VA, USA), UW228-3 (nonmetastatic primary tumour derived), D458 (derived from metastatic cells in the CSF) brain tumour cell lines were a gift from John R. Silber (University of Washington, Seattle, DA, USA), HD-MB03 (metastatic primary-tumour derived) brain tumour cell lines were a gift from Till Milde, D425 (derived from the primary-tumour counterpart of D458) from Marcel Kool (both at DKFZ, Heidelberg, Germany), and ONS76 (metastatic primary-tumour derived) brain tumour cell lines were a gift from Annette Künkele (Charité Universitätsmedizin, Berlin, Germany). FB83 was derived inhouse from human foetal brain tissue as previously described [20]. CHLA-01 and CHLA-01R were cultured in Dulbecco’s Modified Eagle Medium (DMEM) supplemented with 2% B-27, 20 ng/mL Epidermal growth factor (EGF, Gibco, PHG0315), and 20 ng/mL basic fibroblast growth factor (bFGF, Gibco, PHG0266). DAOY, D283, D425 and D458 cells were cultured in DMEM with 10% foetal bovine serum (FBS). FB83 cells were cultured in DMEM with 15% FBS. UW-228-3 cells in DMEM/F-12 with 15% FBS and 1% sodium pyruvate. ONS76 and HD-MB03 cells were cultured in RPMI 1640 with 10% FBS. In all cases, FBS used was (HyClone SH30541.03 (Logan, UT, USA)).

All cell lines were grown under antibiotic-free culture conditions at 37 °C in a humidified atmosphere with 5% CO_2_. Prior to EV isolation, CHLA-01 and CHLA-01R cell lines were grown without EGF and bFGF, and D425 and D458 cell lines were grown in DMEM with 2% exosome-depleted FBS for 48 h. Exosome-depleted FBS was generated by the pelleting of extracellular vesicles by overnight ultracentrifugation at 100,000× *g* at 4 °C. Mycoplasma testing was performed monthly using a PlasmoTest^TM^ mycoplasma detection kit (InvivoGen (San Diego, CA, USA); rep-pt1) as per the manufacturer’s instructions.

### 2.2. PrestoBlue^®^ Assay

To measure changes in cell viability and the proliferation of cells, a PrestoBlue^®^ (ThermoFisher (Waltham, MA, USA), A13262) assay was used. Cells were assayed at a final dilution of 1:10 for 60 min at 37 °C and 5% CO_2_ and fluorescence was measured at 560/590 nm using a FLUOstar Omega (BMG Labtech, (Ortenberg, Germany)) microplate reader.

### 2.3. Isolation of Exosomes

To isolate exosome cell populations, cells were cultured in 10 T-75 flasks up to 30% confluence, washed twice with Hanks’ Balanced Salt Solution (HBSS, Gibco (Loughborough, UK)) and incubated in media supplemented with exosome-depleted FBS for 48 h. Cell culture supernatants were collected and cells were removed by centrifugation for 5 min at 750× *g* at 4 °C. The pellet was discarded and the supernatant was centrifuged for 15 min at 1500× *g* at 4 °C to remove smaller cell debris. To remove larger EVs, the supernatant was then centrifuged for 35 min at 14,000× *g* at 4 °C. The remaining supernatant was filtered through a 0.22 μm filter (Millipore, Burlington, MA, USA) and then centrifuged for 2 h at 100,000× *g* at 4 °C. The pellet was washed by resuspension in phosphate-buffered saline (PBS, Gibco (Loughborough, UK)) and centrifuged again for 2 h at 100,000× *g* at 4 °C.

### 2.4. Migration and Invasion Assays

Cell migration and invasion were quantified in a modified Boyden chamber assay. In both, the lower wells of the chamber were filled with DMEM + 10% FBS (D425 and D458 cells) or EGF, FGF (20 ng/mL) and 2% B-27 (CHLA-01 and CHLA-01R cells) and sealed with a polycarbonate transwell insert with a pore diameter of 8 μm (Greiner Bio-One Greiner (Kremsmünster, Austria 662638)). For the invasion experiments, transwell inserts were coated with Collagen IV (Bio-Techne (Abingdon, UK) 3410-010-02)) diluted to 200 μg/mL in H_2_O and laminin I (Cultrex, 3400-010-02) and diluted to 100 μg/μL in serum-free media. This setting is thought to resemble the in vivo situation of tumour migration through the basement membrane. The 1 × 10^5^ cells were seeded in the upper wells in a medium devoid of FBS. The chamber was incubated for 24 h at 37 °C, 5% CO_2_ and subsequently, the number of invasive cells in the lower chamber was quantified relative to the standard curve. For the exosome treatments, cells were pretreated with the indicated concentrations of exosomes 24 h prior to the seeding of tumour cells in the upper wells.

### 2.5. Generation of Stable Knockdown Cell Lines

Cell lines with stable knockdown of *BSG* and *MMP-2* expression were generated through shRNA-mediated gene silencing using commercialised virus particles. GIPZ™ Lentiviral particle starter kits (Horizon Discovery, (Waterbeach, UK) *BSG*; VGH5526-EG4313, *MMP-2*; VGH5526-EG682) were used for subsequent transduction of the CHLA-01R and D458 cell lines, following the GIPZ™ lentiviral shRNA manual.

### 2.6. Preparation of Cerebrospinal Fluid Samples

Cerebrospinal fluid (CSF) samples were obtained from paediatric patients who had lumbar punctures within a fortnight of the surgical resection of their primary or recurrent/relapsed medulloblastoma at our institution. The samples were twice centrifuged at approximately 100× *g* for 10 min and the resulting supernatant was further centrifuged at 300× *g* for 10 min before storing them at −80 °C. The CSF samples were then defrosted over ice and equal volumes were assayed in gelatin zymography experiments.

### 2.7. Detection of Metalloproteinase Activity by Zymography

The activity of gelatinases MMP-2 and MMP-9 was determined using gelatin zymography. Cell supernatants (15 μL), CSF (15 μL) or exosomes (20 μg) were mixed 1:4 with 4X NuPage LDS sample buffer (11836170001, Roche (Basel, Switzerland)) (without β-mercaptoethanol) and loaded onto 10% SDS-PAGE gels that had been supplemented with 3 mg/mL gelatin (Sigma Aldrich (St. Louis, MO, USA). Recombinant MMP-2 (Sigma Aldrich (St. Louis, MO, USA), 86607) was also added as a reference marker. Electrophoresis was used to separate proteins according to their molecular weight. Subsequently, to restore MMP activity, gels were incubated in zymogram renaturing buffer (Thermo Fisher Scientific (Waltham, MA, USA) at room temperature for 30 min. Renaturing buffer was removed and replaced with a zymogram developing buffer (Thermo Fisher Scientific (Waltham, MA, USA) and incubated overnight at 37 °C, during which time gelatinases degrade the gel. This degradation was visualised by the staining of the gels with 0.5% *w*/*v* Coomassie blue R-250 in 40% *v*/*v* methanol, 10% *v*/*v* glacial acetic acid (Bio-Rad (Hercules, CA, USA) at room temperature for 15 min. To reduce background staining, gels were washed in a destaining solution (40% *v*/*v* methanol, 10% *v*/*v* glacial acetic acid) for two hours to reveal the gelatinase bands. Gels were fixed in a fixative solution (2% paraformaldehyde, 0.075 M lysine, and 0.01 M sodium periodate, pH 7.4) for 15 min before being dried and photographed.

### 2.8. Western Blot

Cells and exosomes were lysed in a lysis buffer (1% Triton-x100, 1 mM EDTA, 150 mM NaCl, 20 mM Tris pH 7.5) supplemented with 1 × cOmplete™ EDTA-free Protease Inhibitor Cocktail (11836170001, Roche (Basel, Switzerland)) and protein concentrations were determined using a Bradford assay (Bio-Rad) in relation to a BSA standard curve. Proteins were separated according to molecular weight by SDS-PAGE and transferred onto a PVDF membrane (Scientific Lab Supplies 10600023). For protein detection, membranes were blocked with 5% nonfat milk and 1% Tween/PBS, and subsequently incubated with primary antibodies against MMP-2 (Abcam, 86607, 1:500), EMMPRIN (Santa Cruz, 46700, 1:500), CD9 (Cell signalling, 13174, 1:1000), Alix (Cell signalling, 2171, 1:1000), Annexin V (Cell signalling, 8555, 1:1000) and Histone 4 (Cell signalling, 41328, 1:1000). Appropriate horseradish peroxidase-conjugated secondary antibodies were applied. GAPDH (Sigma-Aldrich, 406609, 1:2000) was used as the loading control. Protein signals were detected using an enhanced chemiluminescence (ECL) solution (Thermo Fisher Scientific (Waltham, MA, USA); 32106). Chemiluminescence was then measured in the LAS-3000 mini biomolecular imager.

### 2.9. Quantitative Real-Time Polymerase Chain Reaction

RNA isolation was performed using the miRNeasy micro kit (Qiagen (Hilden, Germany)). RNA samples were transcribed into cDNA using reverse transcriptase (Invitrogen (Waltham, MA, USA); 1080-044). Gene expression of the resultant cDNA template was assessed by quantitative reverse transcription PCR (CFX96 real-time PCR machine; BIORAD, (Hercules, CA, USA)) and iQ SYBR SuperMix (BIORAD, (Hercules, CA, USA)). Primer sequences: *MMP-2* forward: 5′ GCCTTTAACTGGAGCAAAAACAA 3′, reverse: 5′ TCCATTTTCTTCTTCACCTCATTG 3′, *BSG* forward: 5′ GTTCTTGCCTTTGTCATTCTG 3′ reverse: 5′ TCACCATCATCTTCATCTACGA 3′. The house-keeping gene *GAPDH* forward: 5′ ATGTTCGTCATGGGTGTAA 3′, reverse: 5′ GTCTTCTGGGTGGCAGTGAT 3′ was used as a control to normalize the data and the relative mRNA expression level was calculated using the ΔCt method.

### 2.10. Transmission Electron Microscopy

To visualize exosomes, vesicle pellets were resuspended in 2% paraformaldehyde and applied to Cu-Rh formvar-coated 200 mesh grids (Agar 53 Scientific, (Los Angeles, CA, USA)) for 3 min. Absorbent paper was used to gently remove any excess suspension. Negative contrast was achieved by incubating the grid for 30 s in 1% uranyl acetate, and excess liquid was blotted off. Subsequent analyses of vesicles were carried out using a JEOL 2100+ transmission electron microscope and iTEM software (Olympus, (Tokyo, Japan)).

### 2.11. NanoFCM

Exosome samples were diluted in PBS (phosphate buffered saline) 1:10–1:100 and analysed using the Flow Nano Analyzer (NanoFCM Inc. (Nottingham, UK)), according to the manufacturer’s protocol [21]. Briefly, lasers were calibrated using 200 nm control beads (NanoFCM Ltd.), which were analysed as a reference for particle concentration. A mixture of various-sized beads (NanoFCM Inc.) was analysed to set a reference for size distribution. PBS was analysed as a background signal. Particle concentrations and size distributions were calculated using NanoFCM software (NanoFCM profession V1.0) and normalised to the cell number and dilution necessary for adequate NanoFCM reading.

### 2.12. Nanoparticle Tracking Analysis

Particle counts and size distributions were determined for each extracellular vesicle preparation using a nanoparticle tracking analysis (NTA) (NanoSight Ltd. (Malvern, UK). The instrument was configured with a 488 nm LM14 laser module and a high-sensitivity digital camera system (OrcaFlash2.8, Hamamatsu C11440, NanoSight Ltd., Amesbury, UK). Prior to the analysis of extracellular vesicle samples, 100 nm standard latex beads were tested as a control to confirm that the NTA measurements were accurate. Samples were diluted in PBS, to a concentration of between 2 × 10^6^ and 5 × 10^7^ particles/mL within the linear range of the instrument. Five replicate videos of 20 s were taken at 25 °C, with samples under controlled flow, and analysed using NTA software (version 2.5), with the minimal expected particle size set to automatic, and camera sensitivity set at 12–16. The detection thresholds were set at 1–3 to reveal small particles.

### 2.13. Statistical Analysis

Results are shown as mean ± SEM of the indicated number of independent experiments. The statistical significance of differences in the group results was compared using a one- or two-way analysis of variance (ANOVA) with multiple comparison testing as indicated. All statistical analyses and plots were carried out using GraphPad Prism 8 (GraphPad Software Inc., La Jolla, CA, USA).

### 2.14. Bioinformatic Analysis of Published Datasets

Published medulloblastoma patient datasets were accessed and analysed using the R2: Genomics Analysis and Visualization Platform (http://r2.amc.nl, (accessed on 3 October 2022)).

## 3. Results

### 3.1. Metastatic Cell Lines Release More Exosomes Than Their Primary Counterparts

The current international consensus recognises four distinct molecular subgroups of medulloblastoma each associated with different patterns of metastasis and overall prognosis—wingless (WNT), sonic hedgehog; (SHH), group three and group four—these can now be further categorised into second-generation subgroups [22,23]. WNT patients classically have a favourable prognosis (>90% 5-year survival); SHH patients display heterogeneous outcomes associated with the age of diagnosis and specific genetics (TP53 mutation status). The underlying biology of group three and group four patients remains less clear, and a substantial number of these patients relapse and are associated with high levels of metastasis [22,23].

Exosomes were isolated from cultured cell lines that represent pairs of metastatic and primary-tumour origins. Since group three and group four medulloblastoma molecular subgroups have the highest occurrence of metastasis at diagnosis [23,24,25] cell lines were used from these subgroups. The two group three cell lines were the primary cell line D425 and the metastatic D458 cell line. The two group four cell lines were the primary CHLA-01 cell line and the metastatic CHLA-01R cell line. These cell lines were valuable in gaining insight into the molecular, genetic, and proteomic signature changes that occurred between the primary-tumour stage and the dissemination to a secondary site.

Exosomes were isolated by ultracentrifugation and characterised according to MISEV criteria [26]. We observed enrichment for the exosome-associated markers CD9, Annexin V, and Alix, and an absence of histone four, a nuclear marker that would indicate contamination with cellular debris (Figure 1Ai,Aii; full-length blots in Appendix A). Exosomes were visualised by transmission electron microscopy (TEM), revealing that medulloblastoma cell lines release a heterogeneous population of “cup-shaped” spherical vesicles (Figure 1B), characteristic of exosomes under TEM [27,28]. There was some heterogeneity in size (diameters varying between 30–150 nm) and appearance of the structures visualized, though little evidence of nonvesicular contamination [29]. Similarly, particle size distribution and particle concentrations, as measured by NTA and NanoFCM (Figure 1C,D), further supported that the isolated particles were indeed exosomes.

A notable feature of cancer cells is that they produce exosomes in greater amounts than normal cells. Several studies have shown that exosome numbers are elevated in the plasma of cancer patients compared to healthy controls; an increase in the levels of extracellular vesicles released correlates with poor prognosis and survival outcomes [30,31]. Exosomes were therefore isolated from a series of medulloblastoma cell lines representing different subgroups and metastatic capacities. Using both particle concentrations and total protein measurements, metastatic cells produce more exosomes relative to primary cells (Figure 1E,F), indicating that they may play a vital role in increasing medulloblastoma metastasis.

### 3.2. Treatment of Medulloblastoma Cells with Migratory-Derived Exosomes Enhances Cell Invasion and Migration

To determine whether metastatic cell-derived exosomes conferred a phenotype upon the recipient cell lines, primary cell lines were cocultured with exosomes isolated from their matched metastatic cell line pair. Migration and invasion of cell lines were measured using a modified Boyden chamber transwell model, recapitulating invasion through the blood–brain barrier (schematic representation Figure 2A). Initially, the migration of cell lines in the absence of added exosomes was measured. When comparing matched cell line pairs, there was a higher level of cell migration in the metastatic cells compared to the primary cells (Appendix A). The effects of exosomes on cell invasion and migration were then assessed by adding exosomes derived from metastatic cells to primary cells in the insert. Interestingly, CHLA-01 cells showed significantly enhanced tumour migration and invasion in response to metastatic exosome addition, compared to the vesicle-free supernatant (Figure 2B). Similarly, D425 cells showed a significant increase in invasion in response to metastatic exosome addition (Figure 2C). Moreover, exosome-free supernatant or PBS had no effect on cell migration rates (Appendix A), confirming that metastatic exosome-induced tumour cell migration and invasion is a specific and not an artefactual effect. In both cell lines tested, there was only a small increase in proliferation after metastatic exosome stimulation (Figure 2D,E), suggesting that the direct impacts on recipient cell migration and invasion are conveyed by exosome stimulation.

Since exosomes have been described as transferring malignant characteristics to surrounding cells, the influence of exosomes on a nonmalignant cell line (FB83) derived from foetal neuronal stem cells was also explored. As shown in Figure 2F, stimulation with metastatic exosomes (derived from CHLA-01R cells) alone was able to induce an invasive phenotype in the FB83 cells (*p* ≤ 0.01), suggesting that recipient cells did not need to be predisposed to an invasive phenotype, or even a cancerous one, prior to exosome stimulation. A significant increase in proliferation was not observed until 72 h (Figure 2G), again indicative of migratory effects only in this cell line.

### 3.3. MMP-2 and EMMPRIN Are Abundant in Proinvasive Exosomes

Having established that metastatic cells were able to release exosomes that endowed an invasive capability on the recipient cells, we then investigated the protease content of our matched lines. In each pair, we were able to identify higher relative expression of MMP-2 in the metastatic line (Appendix A). Matrix metalloproteinases (MMPs) are proteases that have been heavily implicated in the modulation of the TME and demonstrated to be upregulated in the metastasis of several solid tumours [32,33]. Their tight regulation is, in part, mediated by an extracellular matrix metalloproteinase inducer (EMMPRIN), a glycoprotein commonly enriched on the surface of tumour cells and associated with tumour progression and poor patient outcomes [34,35]. EMMPRIN has been shown to mediate MMP release [36] and, more recently, it has been found in tumour-derived extracellular vesicles [37] where it also mediated an MMP-inducing effect [38].

Elevated EMMPRIN expression levels have been correlated with a higher metastatic stage in medulloblastoma [19,39]. Further analysis of large-scale publicly-available datasets shows that when medulloblastoma patients were stratified according to the mRNA levels at diagnosis. Those with higher *BSG* expression had significantly worse five- and ten-year overall survival than patients with lower *BSG* expression (Appendix A), suggesting that the expression level of *BSG* may represent a negative prognostic factor for overall survival in medulloblastoma patients. Moreover, *BSG* gene expression was assessed across a panel of subgrouped medulloblastoma cell lines with known metastatic status. Interestingly, expression was observed to be higher in group three and group four cell lines, the two subgroups commonly associated with poor prognosis and a higher potential for metastatic dissemination (Appendix A). A significant biochemical property of EMMPRIN is that it can appear in diverse glycosylated forms, with large variation in molecular weights. Previous research has suggested that differential glycosylation of EMMPRIN exhibits functional relevance in tumour cells, with the highly glycosylated form being associated with enhanced cell adhesion, cell migration, and MMP-1 and MMP-2 production [36]. We, therefore, aimed to identify the expression patterns and glycosylation forms of EMMPRIN in matched medulloblastoma cell lines. Metastatic cell lines were significantly enriched with highly glycosylated EMMPRIN, compared to the primary cell lines which were distinguished by the low glycosylation variant of EMMPRIN (Appendix A).

Having been shown to correlate with poor patient outcomes and a higher grade in multiple cancers [40], it was postulated that the expression of MMP-2 would represent a poor prognosis factor in medulloblastoma. However, when medulloblastoma patients were stratified according to mRNA levels at diagnosis, those with elevated MMP-2 expression displayed a better five- and ten-year overall survival than patients with low MMP-2 expression (Appendix A). Additionally, when comparing MMP-2 gene expression in matched-primary and metastatic medulloblastoma cell lines there was not a consistent pattern of high MMP-2 in metastatic cells and low MMP-2 expression in the primary cells. However, this “high is worse” pattern was observed at the protein level, indicative that MMP-2 protein levels may be a more physiologically relevant marker of medulloblastoma metastasis (Appendix A).

Since medulloblastoma cell lines exhibit high EMMPRIN and MMP-2 levels and their expression was differentially enriched in metastatic cell lines compared to the primary cell lines, we hypothesised that MMP-2, and its inducer EMMPRIN, could be packaged into exosomes for extracellular release, thus contributing to medulloblastoma invasion and migration. A Western blot analysis of exosomes was therefore performed to assess whether the proinvasive phenotype correlated with increased MMP-2 and EMMPRIN levels. An analysis revealed an enrichment of EMMPRIN and MMP-2 in metastatic-derived exosomes compared to primary-derived exosomes (Figure 3Ai–Aiii, full-length blot in Appendix A). Of note, the analysis of EMMPRIN protein size revealed that only metastatic CHLA-01R exosomes were enriched with highly glycosylated EMMPRIN compared to the low glycosylation variant of EMMPRIN. In contrast, the CHLA-01, D458, and D425 exosomes were distinguished by the low glycosylation variant of EMMPRIN.

Since proinvasive exosomes promote cell migration and invasion, we predicted that this effect is mediated by MMP-2 and/or EMMPRIN on the exosomal surface. To investigate this hypothesis, flow cytometry of the exosomes was carried out. The exosomes isolated from the metastatic D458 cell line clearly resolved into two populations based upon EMMPRIN expression (49.2% EMMPRIN positive compared to 50.7% EMMPRIN negative) whereas only a single, lower expressing, population was observed in the D425-derived exosomes (1.6% EMMPRIN positive compared to 98.3% EMMPRIN negative) (Figure 3B,C). For MMP-2 expression, the difference was more subtle, with exosomes isolated from the metastatic D458 cell lines clearly resolved into two populations based upon MMP-2 expression (56.2% MMP-2 positive compared to 43.8% MMP-2 negative) and a lower population of MMP-2-expressing exosomes was observed in the D425 exosomes (38.1% MMP-2 positive compared to 61.9% MMP-2 negative) (Figure 3D,E). These findings reinforced our hypothesis that EMMPRIN and MMP-2 are enriched on metastatic exosomes and are involved in conveying a proinvasive phenotype.

### 3.4. Stimulation of Medulloblastoma Cells with Migratory-Derived Exosomes Enhances Cell Invasion and Migration

MMPs can either be bound to the cell’s membrane or secreted from the cytosol into the extracellular matrix, where they function as modulators of the extracellular space [40]. MMPs are synthesised as inactive proenzymes, termed zymogens, which require activation prior to being functionally active. Importantly, several MMPs in exosomes have been shown to exhibit proteolytic activities and can directly contribute to the degradation of ECM proteins in the extracellular space [41]. We hypothesised that surface-bound MMPs may degrade the extracellular matrix more effectively, creating a path for tumour cells or exosomes to migrate in the extracellular space. To determine whether the exosomal MMP-2 was catalytically active, gelatin zymography of exosomes suspended in PBS was performed and the functional activity of MMP-2 was determined. In accordance with the Western blot results, the metastatic-derived exosomes demonstrated the highest levels of functional MMP-2 activity compared to the primary exosomes (Figure 4A). The inherent differences in the basal levels of exosome-associated MMP-2 present in metastatic-derived exosomes compared to primary exosomes, further suggests that the MMP-2-associated exosomes promote tumour progression.

Since proinvasive exosomes were found to express EMMPRIN and MMP-2 at high levels, and the role of exosome-bound EMMPRIN for the induction of MMPs is well established [42,43], MMP-2 secretion by medulloblastoma cells after exosome stimulation was explored (schematic representation shown in Figure 4B). After preconditioning with proinvasive exosomes, a time-dependent increase in the levels of functionally active MMP-2 secreted into the media compared to buffer-treated cells was observed (Figure 4C,D).

### 3.5. Knockdown of MMP-2 and EMMPRIN Reduces Exosome-Mediated Migration and Invasion

Given that exosomes from metastatic medulloblastoma cells, carrying higher levels of MMP-2 and EMMPRIN could confer an invasive phenotype on recipient cells, we decided to test the direct contribution of MMP-2 and EMMPRIN by genetic knockdown (Appendix A). In group four, medulloblastoma cells’ knockdown of *BSG* or *MMP-2* at the genetic level resulted in a reduction of the ability of cells to invade a collagen-IV–laminin matrix (Figure 5A). Knockdown of *BSG* or *MMP-2* had little effect on either the distribution or the amounts of exosomes secreted by the corresponding cells (Figure 5B). Exosomes isolated from these cell lines did, however, result in a significant decrease in MMP-2 and EMMPRIN exosomal levels (Figure 5Ci–Ciii).

Primary cell lines were cocultured with exosomes isolated from knockdown metastatic cell lines or exosomes isolated from nontargeting controls. The addition of exosomes does not affect *MMP-2* mRNA levels in recipient cells (Figure 5D). However, a significant reduction in the migration of primary group four cell lines was observed when cells were pretreated with exosomes isolated from knockdown EMMPRIN and MMP-2 cell lines (Figure 5E). In addition, the preconditioning with exosomes isolated from MMP-2 knockdown cell lines also reduced the secretion of MMP-2 in primary medulloblastoma cell lines compared to controls (Figure 5Fi). Since EMMPRIN is known to induce MMP-2 production and secretion, it was slightly surprising that an increase in MMP-2 function was observed when primary medulloblastoma cell lines were preconditioned with exosomes isolated from *BSG* knockdown cell lines, (Figure 5Fii). The incomplete knockdown of EMMPRIN in exosomes may account for this observation (Figure 5Ci–Ciii), thus MMP-2 activation persists to some extent. Together, these results indicate that exosomal EMMPRIN has the ability to initiate MMP-2 secretion in the recipient medulloblastoma cells, while exosomal MMP-2 can further stimulate MMP-2 secretion in a positive feedback loop. Increased levels of secreted MMP-2 would support extracellular-matrix degradation, in turn facilitating tumour cell invasion and metastasis.

Group three primary MB cells showed a similar pattern in reduction when using knockdown exosomes; however, the results were not statistically significant (Appendix A). In accordance with the findings that depletion of MMP-2 or EMMPRIN slightly reduced migration of medulloblastoma cells, these results strongly indicate that exosomal MMP-2 and EMMPRIN facilitate the progression of medulloblastoma. This is particularly apparent for group four medulloblastoma.

### 3.6. CSF Levels of Functionally Active MMP2 Increase at Disease Progression

Having consistently demonstrated higher levels of functionally active MMP-2 are secreted from the group four metastatic MB cell line CHLA-01R in comparison to its primary counterpart, we sought to determine whether this finding would be replicated in an ex-vivo pilot study of cerebrospinal fluid (CSF) derived from paediatric medulloblastoma patients. Their key features are tabulated in Figure 6A. Disease progression was defined as patients presenting with the growth of their residuum (Figure 6B), recurrence or relapse of their medulloblastoma. We explored whether the levels of functional active MMP-2 secreted into CSF could be employed as surrogate markers of disease progression, irrespective of their starting point. The examined CSF samples were derived from four patients, all of whom were male and between the ages of 5 and 20 years old. Molecular subgrouping was unfortunately only available for two of the four patients both of whom were determined to be group four MB. While the size of our pilot study was limited by the availability of the matched CSF samples, we observed increased levels of functionally active MMP-2 in three of four patients, which correlated with disease progression in patients one, three, and four (Figure 6C). Furthermore, the observation of a strong MMP-2 signal in the CSF derived from patients one and four at the point of disease progression, appeared to correlate with prognosis, in contrast to CSF cytology, which was negative in both cases (Figure 6A).

## 4. Discussion

Metastatic dissemination is the predominant cause of the mortality of patients with medulloblastoma [44]. Increasing evidence suggests that exosomes serve as key mediators in tumour metastasis [45]. It has been reported that exosomes from highly metastatic tumour cell lines could significantly enhance the migration capacity of less migratory recipient cells [35]. Similarly, the present study revealed that exosomes isolated from migratory medulloblastoma cell lines directly enhanced the invasive potential of less-migratory matched primary-tumour cells in a heterologous stimulation loop. The effect was specific to exosomes since the exosome-free supernatant had no influence on tumour invasion.

Several studies have demonstrated that exosomes are able to transfer malignant characteristics from highly invasive tumour cells to surrounding nonmalignant cells [46,47]. In line with this, the invasive capacity of nonmalignant foetal neuronal stem cells was enhanced when stimulated with heterologous exosomes derived from highly migratory medulloblastoma cell lines, indicating that recipient cells did not need to be predisposed to an invasive phenotype prior to exosome stimulation.

Although some studies have already demonstrated the tumour-supporting functions of EMMPRIN or MMP-2 in medulloblastoma metastasis [39,48,49,50], the presence of these proteins on medulloblastoma exosomes has not yet been investigated. In this current study, we have advanced the understanding of the mechanism of action of MMP-2 and its inducer EMMPRIN in medulloblastoma progression by demonstrating that MMP-2 and EMMPRIN are associated with secreted exosomes. Importantly, Western blot and flow cytometry analysis revealed that proinvasive metastatic exosomes were significantly enriched with membrane-associated EMMPRIN and MMP-2 compared to the primary exosomes. It is therefore reasonable to propose that exosomal MMP-2 and EMMPRIN are surface-associated in invasive exosomes, as this location is required for a protease to exert its proteolytic activity on extracellular proteins. Recently, it has been suggested that EVs possess a protein corona [51,52], which includes extracellular matrix components such as fibronectin. It would be interesting to see if this protein corona is present in medulloblastoma-derived exosomes and if MMP-2 or EMMPRIN might either be interacting with it or constituting it.

The role of exosomes in ECM degradation has been described recently [50,53]. Activated MMP-2 has a wide range of substrates and can degrade basement-membrane collagen IV, elastin, and several other ECM molecules, including interstitial-collagen types I, II, and III [54], and is therefore associated with ECM remodelling as a prerequisite for cellular invasion and migration [55]. Our finding, that functional activated MMP-2 is carried by exosomes and released by metastatic cell lines into the extracellular space, reveals a possible mechanism by which MMP-2 may facilitate the breakdown of the ECM. Degradation of the ECM would create routes for medulloblastoma cells to directionally invade the surrounding environment, and functional MMP-2 on the external surface of exosomes could allow exosomes to reach long-distance target locations. Moreover, shRNA-mediated knockdown of EMMPRIN or MMP-2 in exosomes decreased the promigratory effect of exosomes on recipient cells.

The association between exosomal EMMPRIN/MMP-2 and ECM degradation was reinforced by zymography assays, demonstrating that secretion of active MMP-2 was activated in primary medulloblastoma cell lines upon coculture with metastatic exosomes enriched with high glycosylated EMMPRIN and MMP-2. Validation of this was provided by a significant reduction in this heterologous stimulation when exosomes were isolated from the metastatic cell lines stably transfected with EMMPRIN or MMP-2 shRNAs. Future research is needed to determine how MMP-2 and EMMPRIN are recruited into and out of the exosomes, as modulation of these mechanisms could potentially prevent or limit the establishment of a premetastatic niche that is conducive to MB dissemination throughout the neuroaxis. Achieving absolute purification or complete isolation of EVs from other entities poses significant challenges. Future proteomic and transcriptomic analyses of the sEVs could serve as a valuable approach to elucidate the molecular mechanisms underlying the metastatic effects of these extracellular vesicles.

Additionally, the positive correlation between the increased secretion of functionally active MMP-2, as detected in the CSF zymography, and their disease progression, may improve our accuracy of disease surveillance and potentially provide a surrogate biomarker of prognosis and response to treatment. However, given our modest sample size, this work would need to be replicated with a larger patient cohort, to gain further insight into the generalisability of our findings.

## 5. Conclusions

Taken together, this work demonstrated that medulloblastoma cells release exosomes into the local tumour microenvironment. This creates a favourable milieu within the neuroaxis, driving medulloblastoma metastasis, through extracellular matrix signalling, via surface-associated proteins. Our findings provide important and unique insights into the pathogenesis of medulloblastoma and highlight the need to explore alternative therapeutic approaches to impair MMP-driven mechanisms of tumour invasion and migration. 

## Figures and Tables

**Figure 1 cancers-15-02601-f001:**
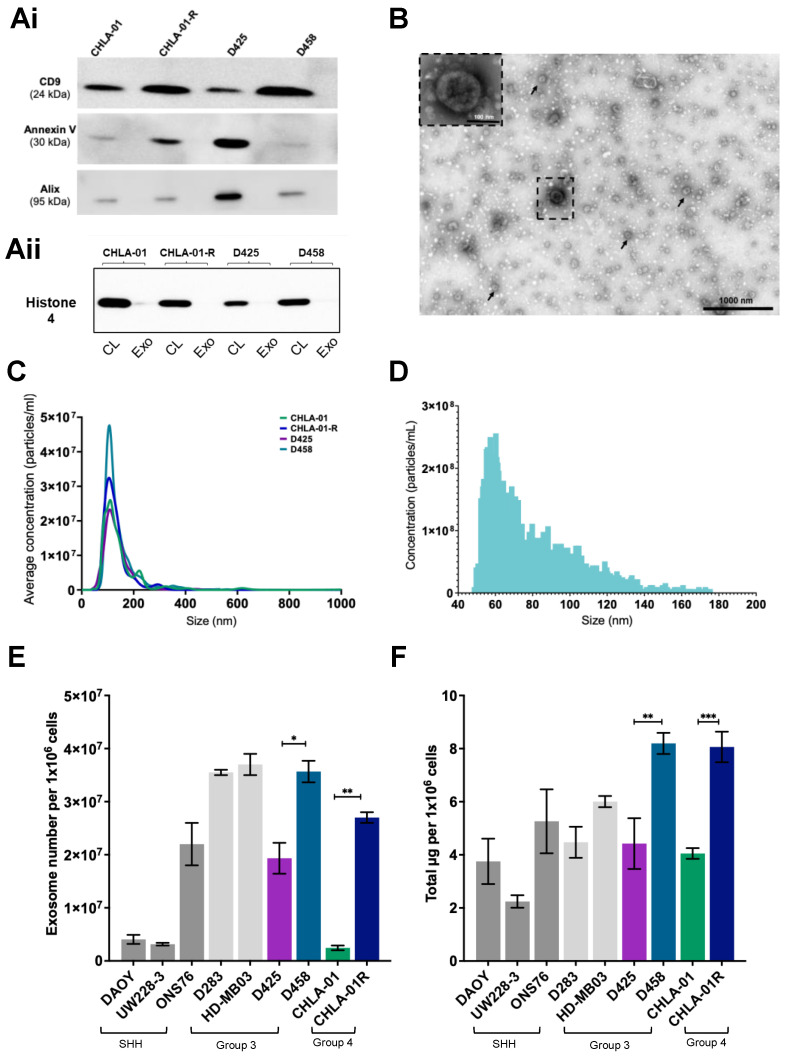
Isolation and characterisation of exosomes from medulloblastoma cell lines. Western blot confirmation that exosomes (**Ai**) express the marker proteins CD9, Annexin V, and Alix, and show no cross contamination with the nuclear protein histone 4 which is only detected in cell lysate (CL) rather than exosome (Exo) fractions (**Aii**). The uncropped blots are shown in Appendix A. (**B**) Exosomes from CHLA-01 cells were imaged by transmission electron microscopy (TEM) and identified as multiple cup-shaped structures ranging from 30–150 nm in size (arrowheads). Scale bar 1000 nm. (**C**,**D**) Exosomal particle concentrations were measured by NTA (**C**) and NanoFCM (**D**). NTA data represent the average of at least three independent repeats. Representative NanoFCM data is shown for CHLA-01-derived exosomes. (**E**,**F**) Metastatic cell lines (D283, HD-MB03, D458, and CHLA-01-R) typically secrete more exosomes than the primary cell lines as determined by exosomal number (**E**) and exosomal protein content (**F**), both corrected for cell density. SHH (sonic hedgehog). Significant differences were calculated using one-way ANOVA analyses with Sidak’s multiple-comparisons test, (* *p* ≤ 0.05, ** *p* ≤ 0.01, *** *p* ≤ 0.005). Data represent the average of three independent experiments with error bars indicating the standard error of the mean (SEM).

**Figure 2 cancers-15-02601-f002:**
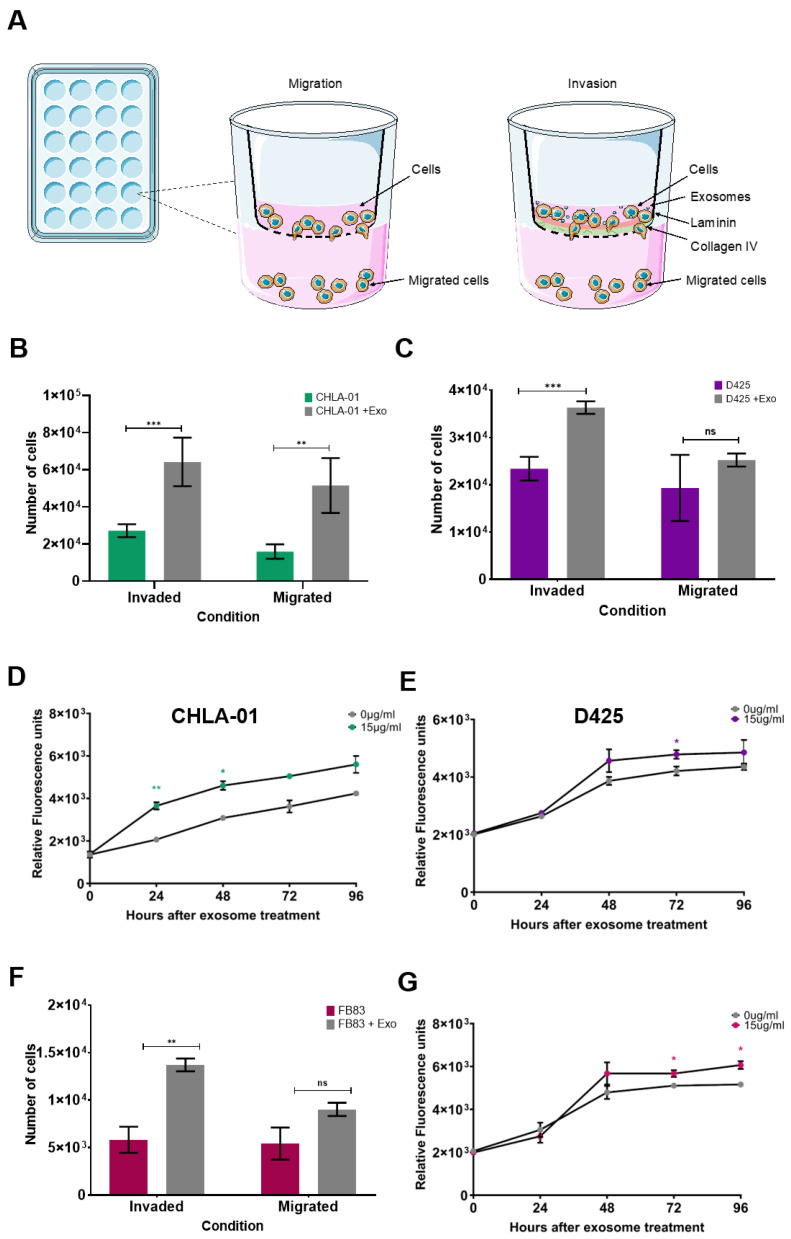
Exosomes from metastatic cell lines can confer a proinvasive phenotype on recipient cells. (**A**) Experimental setup for migration through an uncoated chamber insert, or invasion and migration through a collagen IV- and laminin I-coated insert. (**B**,**C**) Exosomes derived from the metastatic cell lines CHLA-01R and D458 were applied (+Exo) to matched primary cell lines CHLA-01 (**B**) and D425 (**C**), respectively, and led to an increase in cell invasion behaviour (grey bars) compared to exosome-free supernatant control cells (coloured bars). (**D**,**E**) Exosomes from metastatic cell lines conferred only a modest effect on cell proliferation determined by Presto Blue viability assays. (**F**,**G**) Exosomes from the metastatic CHLA-01R cell line were able to confer an invasive phenotype on the noncancerous FB83 cell line (**F**), independent of an effect on cell proliferation (**G**). Significant differences in migration were calculated using one-way ANOVA analyses with Dunnett’s multiple-comparisons post hoc test, (* *p* ≤ 0.05, ** *p* ≤ 0.01, *** *p* ≤ 0.005) (ns = not significant). Data represent the average of three independent experiments with error bars indicating the standard error of the mean (SEM).

**Figure 3 cancers-15-02601-f003:**
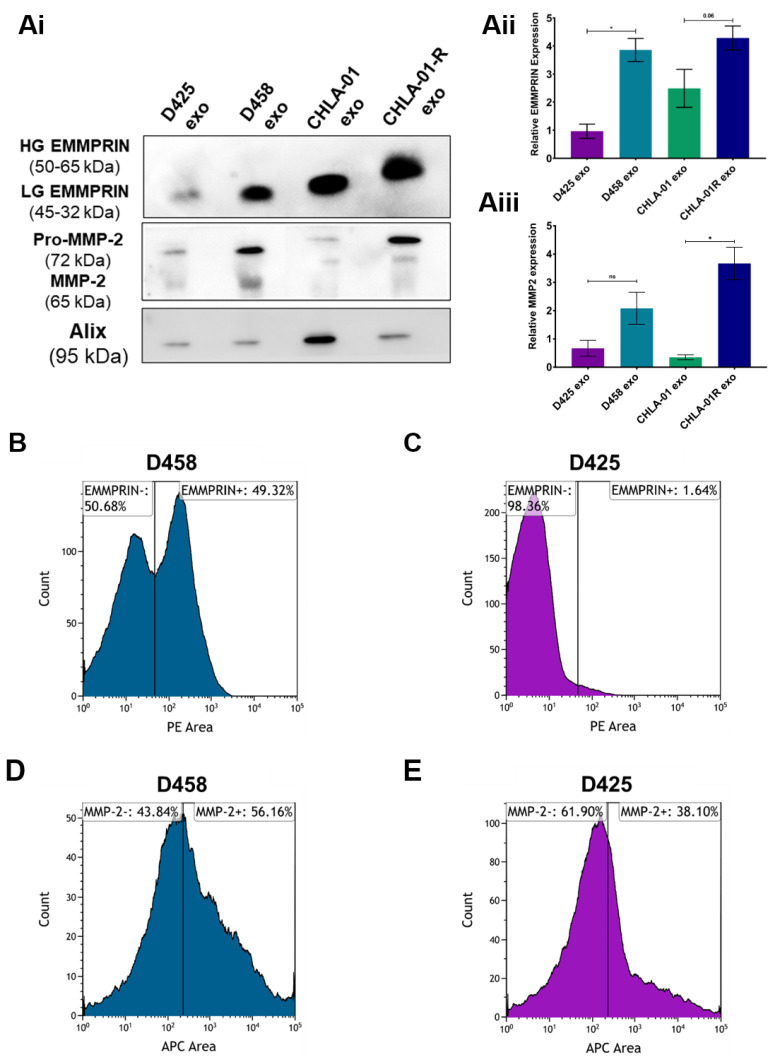
MMP-2 and EMMPRIN are expressed on exosomes released by medulloblastoma cell lines. Exosomal EMMPRIN and MMP2 protein expression was determined by Western blotting (representative image in panel (**Ai**) and was quantified relative to Alix protein expression (**Aii**,**Aiii**). (**B**). Flow-cytometry analysis of EMMPRIN (**B**,**C**) and MMP-2 (**D**,**E**) in medulloblastoma-derived exosomes. Exosomes from the metastatic and primary matched pair of cell lines (D458 and D425) were labelled with anti-EMMPRIN or anti-MMP-2 antibodies and secondary antibodies conjugated to either PE (EMMPRIN) or APC (MMP-2). Gating was identical across cell lines and the percentage of cells in populations representing high and low expression is shown. One representative experiment out of two performed with similar results is shown. Data in (**Ai**–**Aiii**) represent the average of three independent experiments with error bars indicating SEM. Significant differences in protein expression between exosomes derived from the matched cell lines were calculated using the Kruskal–Wallis test with Dunn’s multiple-comparisons post hoc test (* *p* ≤ 0.05, ns = not significant). The uncropped blots are shown in Appendix A.

**Figure 4 cancers-15-02601-f004:**
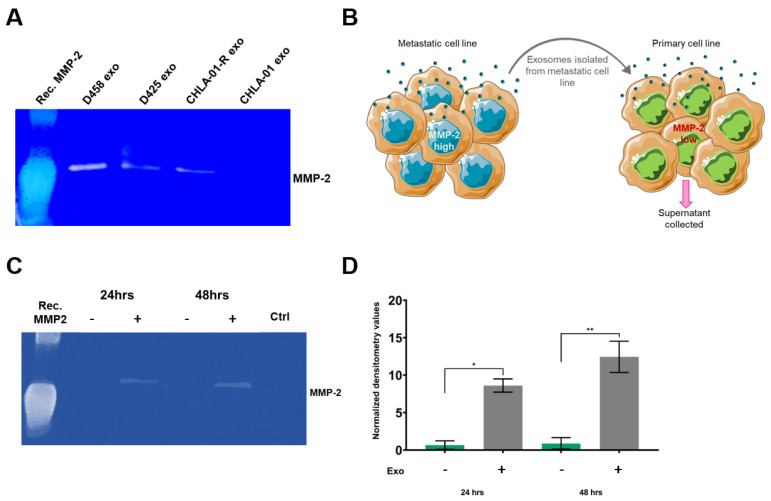
Exosomes contain functionally active MMP-2 and can transfer gelatinase activity onto recipient cell lines. (**A**) Gelatin zymography was used for the detection of MMP-2 activity in medulloblastoma-derived exosomes. The 5 μg of exosomes suspended in PBS were loaded onto gelatin-incorporated zymography gels and the functional activity of the gelatinase and MMP-2 was determined. Data are representative of three independent experiments. (**B**) Schematic representation of the stimulation of a low MMP-2 expressing cell line (CHLA-01) with 10 μg of exosomes from a high MMP-2 expressing cell lines (CHLA-01R). After 24 and 48 h, supernatant from the recipient cell line was collected and loaded onto the gelatin-containing gel. Proteolysis was detected as a white band. MMP-2 levels shown in (**C**) were quantified and displayed graphically in (**D**). Data represent the average of two independent experiments with error bars indicating the standard error of the mean (SEM). Significant differences were calculated using one-way ANOVA analyses with Dunnett’s multiple-comparisons post hoc test, (* *p* ≤ 0.05, ** *p* ≤ 0.01). The uncropped blots are shown in Appendix A.

**Figure 5 cancers-15-02601-f005:**
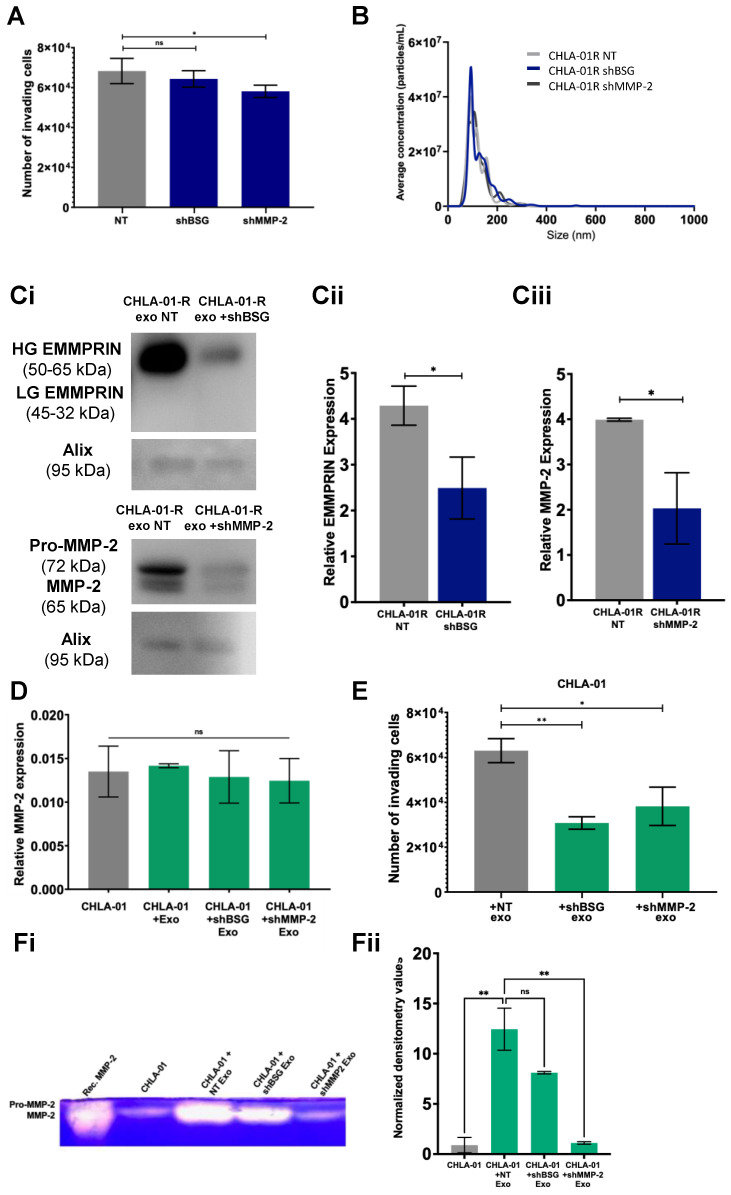
Knockdown of MMP-2 and EMMPRIN appears to reduce exosome-mediated migration and invasion in group four medulloblastoma cell lines. (**A**) Group four cell invasion through a collagen and laminin IV coated transwell chamber insert was quantified by PrestoBlue metabolic staining and compared to cell lines transduced with nontargeting (NT). (**B**) Exosomes from the CHLA-01R knockdown cell lines and nontransduced cells were isolated and their size and concentration were measured by NanoFCM. (**Ci**) Western blot analysis and concurrent densitometry revealed both EMMPRIN (**Cii**) and MMP-2 (**Ciii**) protein levels to be significantly depleted in exosomes isolated from the knockdown cell lines. Densitometry data are presented relative to the Alix loading control and compared to the appropriate nonsilencing control cell line. (**D**) Exosomes from the CHLA-01R knockdown cell lines were applied to the matched parental cell line CHLA-01 (+NT exo: nontargetting, +shBSG exo, knockdown *BSG,* or shMMP-2 exo knockdown MMP-2) following which MMP-2 mRNA expression levels in recipient cells was determined by qRT-PCR. (**E**) The ability of these recipient cell lines to invade through a laminin and collage IV matrix was determined by metabolic assay. MMP-2 activity in the medium of CHLA-01 cells receiving exosomes was determined by gelatin zymography. Proteolysis was detected as a white band. MMP-2 levels shown in (**Fi**) were quantified and displayed graphically in (**Fii**). Significance was assessed by ordinary one-way ANOVA analysis with Sidak’s multiple-comparison tests (* *p* ≤ 0.05, ** *p* ≤ 0.01) (ns = not significant). Data represent the average of at least two independent experiments with error bars indicating the standard error of the mean (SEM). The uncropped blots are shown in Appendix A.

**Figure 6 cancers-15-02601-f006:**
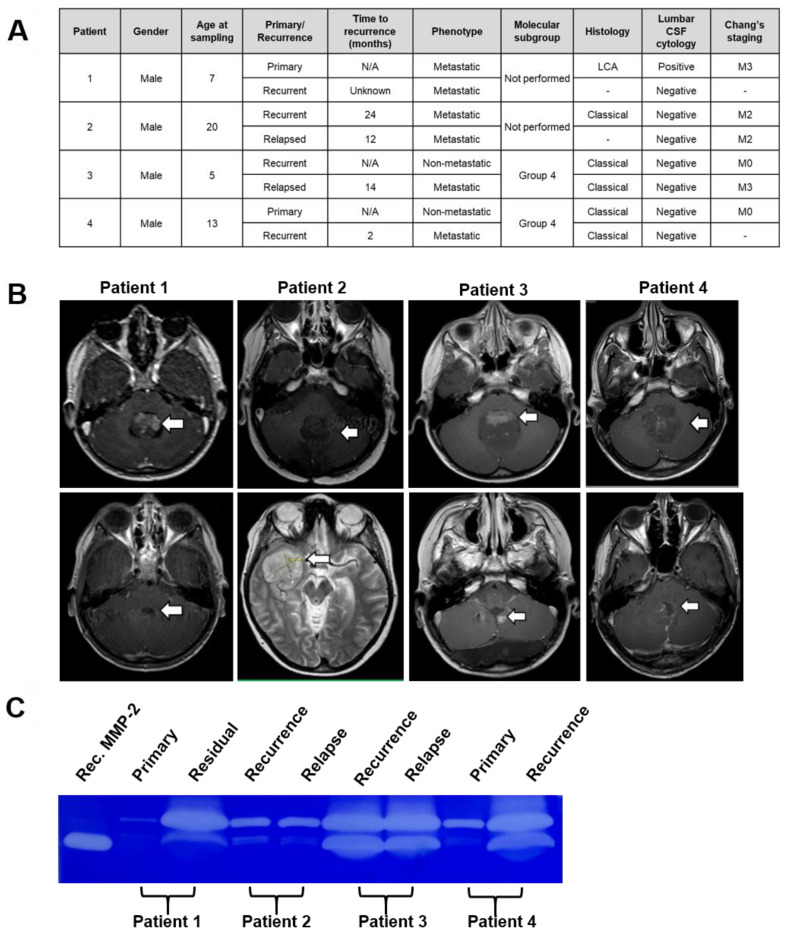
Determination of MMP-2 activity in clinical CSF samples from medulloblastoma patients indicates MMP-2 activity as a possible marker of disease progression. (**A**) Clinical characteristics of 4 medulloblastoma patients. (**B**) Axial MRI with gadolinium T1 and T2 images taken at presentation (top row) and at disease progression (bottom row) with tumours highlighted with white arrows. (**C**) Paired CSF sampled from the patients was examined and the levels of functional activity of MMP-2 determined and normalised to recombinant MMP-2, at their presentation and disease progression. The uncropped blots are shown in Appendix A.

## Data Availability

Data is contained within the article or Appendix A.

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
