# Peer review of "Extracellular Vesicles Potentiate Medulloblastoma Metastasis in an EMMPRIN and MMP-2 Dependent Manner"

_cancers, 2023, doi:10.3390/cancers15092601_

Round 1
Reviewer 1 Report
Review Report for the Manuscript “Extracellular Vesicles Potentiate Medulloblastoma Metastasis in an EMMPRIN and MMP-2 Dependent Manner”
Rating the Manuscript
Originality/Novelty: Is the question original and well defined? Do the results provide an advance in current knowledge?
Yes, in the manuscript the authors focus on whether sEVs, exosomes, mediate communication between medulloblastoma cells and their surroundings to drive metastasis.
Significance: Are the results interpreted appropriately? Are they significant? Are all conclusions justified and supported by the results? Are hypotheses and speculations carefully identified as such?
Yes, the results are interpreted well.
Quality of Presentation: Is the article written in an appropriate way? Are the data and analyses presented appropriately? Are the highest standards for presentation of the results used?
Yes, the article is written well. Data representation, figures and figure captions are very informative.
Scientific Soundness: is the study correctly designed and technically sound? Are the analyses performed with the highest technical standards? Are the data robust enough to draw the conclusions? Are the methods, tools, software, and reagents described with sufficient details to allow another researcher to reproduce the results?
Yes, the data is robust enough to draw conclusions and the methods, tools and methods used in the data analysis are explained properly.
Overall Merit: Is there an overall benefit to publishing this work? Does the work provide an advance towards the current knowledge? Do the authors have addressed an important longstanding question with smart experiments?
Yes. This study provides an advancement to the current knowledge.
English Level: Is the English language appropriate and understandable?
Yes, English language in the manuscript is appropriate and understandable.
Overall Recommendation: This is a well written paper and I only have few comments. Accept after Minor Revisions
Given below are the comments for each section of the manuscript.
Supplementary material and original figures are the same. Are they supposed to contain different figures.
Simple Summary
Line 18: “Metastatic exosomes were shown to potentiate medulloblastoma migration via the active protease, MMP-2, on their surface, resulting in degradation of the extracellular matrix (ECM) creating routes for medulloblastoma cells to invade into the surrounding environment.”
I think it’s better if the authors define the term MMP-2 when it first appears in the manuscript.
Line 21: “Knockdown of MMP-2 and its activator EMMPRIN reduced this invasive potential.”
Please define the term EMMPRIN.
Abstract
The abstract is written and summarizes the content of the manuscript.
Graphical Abstract
Graphical abstract is a good representation of what’s discussed in the manuscript.
Introduction
Please correct the spellings of the word, “nanometer” in line 59.
It’s better if the authors can briefly mention about types of EVs. Also, they should briefly discuss about the size distribution and biogenesis of small EVs and exosomes since they are only focusing on these.
I think authors could add more references in the introduction. Authors need to add references to the following statements.
Line 58: “The most investigated are the small EVs (sEVs or exosomes), which are nanometre-sized vesicles secreted by all cell types and able to cross the blood-brain-barrier (BBB).”
Line 62:” Exosomes represent a unique form of information delivery operating at short and long distances. Tumour-derived exosomes can transfer signals and convey information from tumours to distant tissues and organs.”
Materials and Methods:
2.1 Cell culture
Line 110: “Prior to EV isolation, CHLA-01 and CHLA-01R cell lines 110 were grown without EGF and bFGF, and D425 and D458 cell lines were grown in DMEM 111 with 2% exosome-depleted FBS for 48 hours”.
How do you know 48 hours is enough get FBS-EV free sample? Has this been reported previously?
2.4. Migration and Invasion Assays
Line 134: “In both, the lower wells of the chamber were filled with DMEM +10% FBS (D425 and D458 134 cells) or EGF, FGF (20 ng/mL) and 2% B-27 (CHLA-01 and CHLA-01R cells) and sealed 135 with a polycarbonate transwell insert with pore diameter of 8 μm (Greiner Bio-One 136 Greiner (Kremsmünster, Austria 662638)).”
Please define the terms EGF and FGF.
3. Results
3.1. Metastatic Cell Lines Release More Exosomes Than Their Primary Counterparts
Did you observe any morphological differences between exosomes isolated from Metastatic Cell Lines compared to exosomes isolated from their Primary Counterparts?
3.2. Treatment of Medulloblastoma Cells with Migratory-Derived Exosomes Enhances Cell 299 Invasion and Migration
Line 321: “As shown in Figure 2F, stimulation with 321 metastatic exosomes (derived from CHLA-01R cells) alone was able to induce an invasive 322 phenotype in the FB83 cells (p≤0.01), suggesting that recipient cells did not need to be 323 predisposed to an invasive phenotype or even cancerous, prior to exosome stimulation.”
Please mention the CI whenever you mention a p value.
Figures
All the figures and figure captions are in good quality and easy to follow.
References:
Some of the references are more than 10 years old. It they don’t contain important information authors could replace these with new references.
References: 4,9,18,19,23,25,28,29,31,35,42,43,44,45 and 52
Reviewer 2 Report
The title of the study needs to be visited again and rephrased in a way that helps the reader to understand the meaning of the study.
The introduction contains good information. However, Justification for the importance of the study was not sufficient. Perhaps if they add studies related to the importance of exosomes in studying the metastasis of tumors in general and brain tumors in particular.
When mentioning cells in the Methods section, I prefer to mention the source of the cells and tissue, whether it is the brain or fibroblasts, and so on. Not enough to mention the cell line codes.
The ultracentrifugation method is a standard method for exosome isolation but presents low purity. Why authors did not go for more precise methods.
The authors indicated that there was an increased number of exosomes released from cancer cells than from normal cells. Is the behavior definitely related to the cancer cell or is it due to an excessive increase in the division of cancer cells.
There is a need to reduce the discussion and move it from the results section to the discussion and conclusions section.
How can we be sure that the proteins we are testing are present in the exosomes? Especially since the method of isolating the exosomes may carry with it contamination and particles similar to the exosome in size
Reviewer 3 Report
Thank you very much for the opportunity to review the manuscript. In their paper, British authors assessed the role of EVs-related EMPPRIN and MMP-2 in medulloblastoma metastasis.
Overall, the manuscript is prepared very carefully.
I have no major criticisms. The introduction is written correctly. The methodology used is correct and described in detail. The results are presented accurately.
My comments are as follows:
1. please consider changing exosomes to small EVs,
2. the discussion should include the limitations of the study,
3. references could be supplemented:
doi: 10.3390/cells11182913
doi: 10.3390/ijms21124463
